# TEMPORAL MOTIF MESSAGE PASSING NETWORK FOR DYNAMIC GRAPH REPRESENTATION LEARNING

## ABSTRACT

Although exsisting continuous-time dynamic graph representation learning methods have achieved great success by constructing extra features based on temporal motifs, they fail to fully leverage the spatial and temporal structures of temporal motifs to model the high-order correlations among nodes. An intuitive and innovative approach is to incorporate temporal motifs into message passing networks, which encounters the following challenges: (1) *diverse semantics exhibit in a temporal motif*; (2) *complex high-order correlations exist among the constituent edges of temporal motifs*. To this end, we propose a **T**emporal **M**otif **M**essage **P**assing network (TMMP) for dynamic graph representation learning. Specifically, a temporal motif extension module is proposed to obtain extended temporal motifs based on different time window lengths. Then, an attentive temporal motif encoder is proposed to capture the diverse semantics of temporal motifs. In addition, a dual hypergraph temporal motif message passing mechanism is proposed to comprehensively model the complex relationships among the constituent edges. Extensive experiments on four real-world datasets demonstrate that TMMP achieves the state-of-the-art performance, surpassing the best baseline methods in both AP and AUC metrics on almost all datasets.

## 1 INTRODUCTION

Dynamic graphs employ nodes and timestamped egdes to model complex dynamic systems in the real world, e.g., social networks (Song et al., 2019), e-commerce systems (Kumar et al., 2019), and news event networks (Chen & Chen, 2024). The fundamental objective of dynamic graph representation learning is to embed the target nodes into a low-dimensional vector space and simultaneously preserve both spatial and temporal characteristics of nodes for downstream tasks.

Dynamic graph representation learning methods have attracted growing research interest in recent years, which can be categoried as discrete-time dynamic graph (DTDG) representation learning methods (Pareja et al., 2020; Sankar et al., 2020) and continuous-time dynamic graph (CTDG) representation learning methods (Kumar et al., 2019; Xu et al., 2020; Rossi et al., 2020). DTDG representation learning methods process dynamic graphs as snapshots sequences and separately process the spatial and temporal information, while CTDG representation learning methods leverage timestamped edge sequences to represent dynamic graphs and jointly model both spatial and temporal dimensions.

Despite the advancements of CTDG representation learning methods, most of them neglect the complex spatial and temporal structures in dynamic graphs. To address this limitation, some CTDG representation learning methods have turned to temporal motifs, which are complex subgraph structures effectively revealing interaction patterns in dynamic graphs. Specifically, through occurrence counting of nodes or edges in the instances of temporal motifs, these methods construct extra features based on temporal motifs to better capture the complex spatial and temporal structures in dynamic graphs (Chen et al., 2021; Wang et al., 2021b).

Nevertheless, the aforementioned CTDG representation learning methods fail to fully leverage the spatial and temporal structures of temporal motifs to model the high-order correlations, i.e., simultaneous correlations among three or more nodes. An intuitive and innovative approach is to incorporate temporal motifs into message passing networks. By definition, temporal motifs consist of three or more nodes, leading them intrinsically suited for modeling the high-order correlations.

Moreover, the spatial and temporal structures of temporal motifs can naturally guide the direction of message propagation in the message passing networks. However, this is a non-trivial task and encounters the following challenges. Firstly, *diverse semantics exhibit in a temporal motif.* A given motif may carry distinct semantics under different time window lengths. For example, in financial networks, a cyclic motif like "AB-BC-CA" with a short time window could indicate suspicious activities, e.g., money laundering operations, whereas the same motif with a long time window could reflect legitimate capital recycling. Secondly, *complex high-order correlations exist among the constituent edges of temporal motifs.* These edges maintain both explicit high-order correlations, e.g., co-occurrence in the same motif, and implicit high-order correlations.

To overcome the aforementioned challenges, we propose a **T**emporal **M**otif **M**essage **P**assing network (TMMP) for dynamic graph representation learning. To the best of our knowledge, TMMP is the first work that incorporates temporal motifs into message passing networks, which can model the high-order correlations among nodes. Our main contributions can be summerized as follows:

- We propose a temporal motif extension module to obtain extended temporal motifs based on different time window lengths. In addition, an attentive temporal motif encoder is introduced to encode and attentively aggregate extended temporal motifs guided by the relationships between nodes and extended temporal motifs, which can capture the diverse semantics of temporal motifs.

- We propose a dual hypergraph temporal motif message passing mechanism that employs a heuristic method for explicit hypergraph construction and hypergraph learning for implicit hypergraph construction, which can comprehensively model the complex high-order correlations among the constituent edges of temporal motifs.

- We validate TMMP on four real-world datasets. Experimental results demonstrate that TMMP achieves the state-of-the-art performance, surpassing the best baseline methods in both AP and AUC metrics on almost all datasets.

## 2 RELATED WORK

**Dynamic Graph Representation Learning Method.** Existing dynamic graph representation learning methods primarily fall into two main classes: DTDG and CTDG representation learning methods. DTDG representation learning methods discretize dynamic graphs into sequences of snapshots at predefined time intervals (Zhu et al., 2016; Zhang et al., 2018; Singer et al., 2019; Pareja et al., 2020; Sankar et al., 2020). These methods typically apply static graph representation learning methods on each snapshot and then capture temporal patterns from snapshot sequences. CTDG representation learning methods represent dynamic graphs as timestamped edge sequences (Nguyen et al., 2018; Kumar et al., 2019; Trivedi et al., 2019; Xu et al., 2020; Rossi et al., 2020; Wang et al., 2021b; Souza et al., 2022; Zou et al., 2024). These methods enable direct learning of node representations from edges with more temporal information. Recently, CTDG representation learning methods have turned to extra features. Yu et al. (2023) propose neighbor co-occurrence encodings to capture the correlations between the historical neighbors of target nodes. Cheng et al. (2024) generate GPU-accelerated long-term and short-term structural encodings to capture temporal patterns. Moreover, some of them introduce temporal motifs to capture the complex spatial and temporal structures. Wang et al. (2021b) utilize temporal random walks to search instances of temporal motifs and constructs relative node identities for nodes in walks. Chen et al. (2021) develop temporal motif-enhanced edge features combined with a bicomponent neighbor aggregation method to extract historical and current neighbor information. Nevertheless, they fail to fully leverage the spatial and temporal structures of temporal motifs to model the high-order correlations. Therefore, we propose TMMP for dynamic graph representation learning, which is the first work that incorporates temporal motifs into message passing networks to model the high-order correlations among nodes.

**Hypergraph Neural Network.** Hypergraph neural networks are generalized graph neural networks, specialized in modeling the high-order correlations in many fields, e.g., knowledge graph reasoning (Tang et al., 2024), time series analysis (Shang & Chen, 2024; Shang et al., 2024), and social network analysis (Yang et al., 2020). For example, Tang et al. (2024) introduce two hypergraphs including entity hypergraph and relation hypergraph to model the high-order correlations among entities and relations in temporal knowledge graphs. Shang & Chen (2024) build intra-scale, inter-scale and

mixed-scale hypergraphs to model the high-order interactions among temporal patterns in multi-scale time series data. Considering the outstanding capability of hypergraph neural networks in modeling the high-order correlations, we introduce this framework to model the complex high-order correlations among the constituent edges of temporal motifs. Specifically, a dual hypergraph temporal motif message passing mechanism is proposed to comprehensively model the complex relationships among the constituent edges.

## 3 PRELIMINARIES

**Definition 1 (CTDG)**. A CTDG can be represented as a chronological timestamped edge sequence $G^\tau = \{e_1, e_2, \ldots, e_n\}$, where $n$ denotes the number of edges in the CTDG until timestamp $\tau$. Each edge $e_i = (u_i, v_i, t_i)$ indicates there is an interaction between node $u_i$ and $v_i$ at timestamp $t_i$, where $1 \le i \le n$ and $t_1 \le t_2 \le \cdots \le t_n \le \tau$.

**Definition 2 (Dynamic Graph Representation Learning)**. Dynamic graph representation learning aims to learn a mapping function $g$ to project nodes into a low-dimensional vector space $\mathbb{R}^d$, which can model the spatial structures and temporal evolution of dynamic graphs. Given an edge $e = (u, v, t)$ and a dynamic graph $G^\tau$, the mapping function $g$ projects node $u$ and $v$ into low-dimensional representations $\boldsymbol{h}_u^t \in \mathbb{R}^d$ and $\boldsymbol{h}_v^t \in \mathbb{R}^d$.

**Definition 3 (Temporal Motif)**. A temporal motif can be represented as a chronological timestamped edge sequence where edges occur within a specified time window. Formally, A temporal motif with a time window of length $\delta$ is defined as $m = \{e_1, e_2, \cdots, e_{\mathcal{N}_m}\}$, where $\mathcal{N}_m$ denotes the number of edges in the temporal motif and $t_1 < t_2 < \cdots < t_{\mathcal{N}_m}$, $t_{\mathcal{N}_m} - t_1 \le \delta$.

**Definition 4 (Temporal Motif Instance)**. Given a temporal motif $m = \{e_1, e_2, \ldots, e_{\mathcal{N}_m}\}$ with a time window of length $\delta$, a dynamic subgraph $\widehat{m} = \{e_1', e_2', \ldots, e_{\mathcal{N}_m}'\} \subset G^\tau$ is an instance of $m$ if it satisfies the following conditions: (1) $m$ and $\widehat{m}$ share the same spatial and temporal structure; (2) all the edges of $\widehat{m}$ occur within the time window of length $\delta$.

## 4 METHODOLOGY

### 4.1 OVERVIEW

The framework of our TMMP is shown in Figure 1. Given an edge $(u, v, t)$, a subgraph encoding module extracts $\mathcal{N}_S$ historical edges of $u$ and $v$ before timestamp $t$ to construct subgraph $G_u^t$ and $G_v^t$. Next, a temporal motif extension module extends original temporal motifs into extended temporal motifs based on different time window lengths. An attentive temporal motif encoder is introduced to encode and attentively aggregate extended temporal motifs. Then, a dual hypergraph temporal motif message passing mechanism is introduced to perform message passing on the explicit hypergraph and implicit hypergraph, which are constructed by a heuristic method and hypergraph learning, respectively. Finally, a Transfomer encoder is used to derive the final representations of $u$ and $v$ at timestamp $t$, which are aware of high-order correlations among nodes in subgraphs. Link prediction is performed based on the final representations.

### 4.2 SUBGRAPH ENCODING MODULE

**Subgraph Construction.** Subgraphs can reveal the spatial and temporal characteristics of target nodes. By modeling the high-order correlations among nodes in subgraphs, we can obtain more expressive representations of target nodes. For clarity in following subsections, given an edge $(u, v, t)$, we describe the operations only for target node $u$, with analogous procedures applied to target node $v$. Subgraph $G_u^t$ is constructed based on $\mathcal{N}_S$ historical edges of $u$ before timestamp $t$.

**Node/Edge Feature.** Nodes and edges in $G_u^t$ possess meaningful features to help model the high-order correlations, which are denoted by $\boldsymbol{X}_{u,\mathrm{N}}^t \in \mathbb{R}^{\mathcal{N}_S \times d_\mathrm{N}}$ and $\boldsymbol{X}_{u,\mathrm{E}}^t \in \mathbb{R}^{\mathcal{N}_S \times d_\mathrm{E}}$, respectively.

**Time Encoding.** Following Xu et al. (2020), given $x$ in $G_u^t$, we encode the time interval $\Delta t_x = t - t_x$ between $u$ and $x$ as follows:

$$\boldsymbol{\Phi}(\Delta t_x) = \sqrt{1/d_\Phi}[\cos(\omega_1 \Delta t_x), \sin(\omega_1 \Delta t_x), \ldots, \cos(\omega_{d_\Phi} \Delta t_x), \sin(\omega_{d_\Phi} \Delta t_x)] \in \mathbb{R}^{d_\Phi}, \quad (1)$$

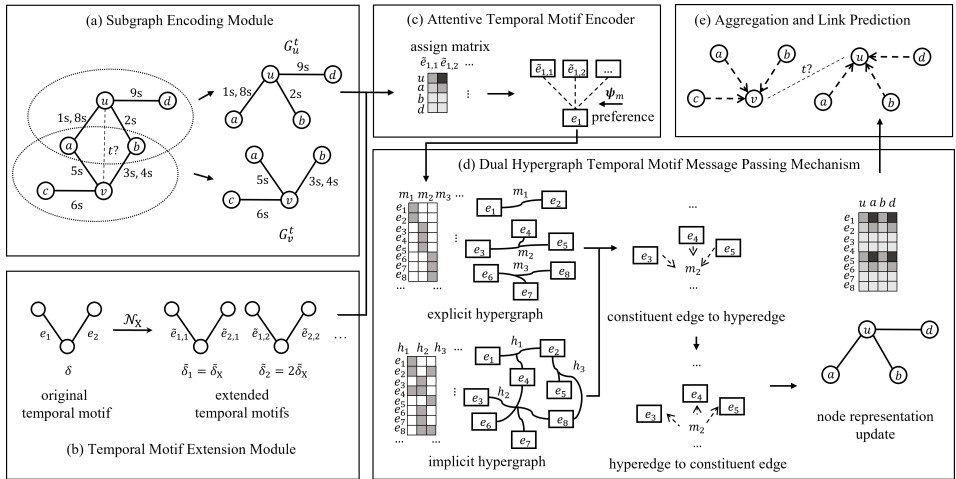

Figure 1: The framework of TMMP.

where $\omega_1, \ldots, \omega_{d_\Phi} \in \mathbb{R}^{d_\Phi}$ are trainable parameters and $d_\Phi$ is the time encoding dimension. The time encodings of nodes in $G_u^t$ is denoted by $\boldsymbol{X}_{u,\Phi}^t \in \mathbb{R}^{\mathcal{N}_\mathrm{S} \times d_\Phi}$.

**Subgraph Co-occurrence Encoding.** To take into account the relationships between $G_u^t$ and $G_v^t$, we follow Yu et al. (2023) to encode the occurrence countings in both $G_u^t$ and $G_v^t$ of nodes. Supposing $G_u^t$ contains $\{a, b, a, d\}$ and $G_v^t$ contains $\{b, b, a, c\}$, the occurrence countings $\boldsymbol{c}_a^t = [2, 1]$, $\boldsymbol{c}_b^t = [1, 2]$, $\boldsymbol{c}_c^t = [0, 1]$, and $\boldsymbol{c}_d^t = [1, 0]$. Therefore, the occurrence countings of $G_u^t$ and $G_v^t$ are denoted by $\boldsymbol{C}_u^t = [[2, 1], [1, 2], [2, 1], [1, 0]]^\mathrm{T}$ and $\boldsymbol{C}_v^t = [[1, 2], [1, 2], [2, 1], [0, 1]]$, respectively. Given $\boldsymbol{C}_u^t \in \mathbb{R}^{\mathcal{N}_\mathrm{S} \times 2}$, we encode the occurrence countings to obtain subgraph co-occurrence encodings as follows:

$$\boldsymbol{X}_{u,\mathrm{C}}^t = \mathrm{MLP}_\mathrm{C}(\boldsymbol{C}_u^t[:, 0]) + \mathrm{MLP}_\mathrm{C}(\boldsymbol{C}_u^t[:, 1]) \in \mathbb{R}^{\mathcal{N}_\mathrm{S} \times d_\mathrm{C}}, \tag{2}$$

where $\mathrm{MLP}_\mathrm{C}(\cdot)$ is an MLP encoder for subgraph co-occurrence encodings.

**Node Representation Initialization.** To initialize the representations of nodes in $G_u^t$, we align node features, edge features, time encodings, and subgraph co-occurrence encodings to the same dimension and concatenate them together, which can be formulated as follows:

$$\widetilde{\boldsymbol{Z}}_{u,*}^t = \boldsymbol{X}_{u,*}^t \boldsymbol{W}_* + \boldsymbol{b}_* \in \mathbb{R}^{\mathcal{N}_\mathrm{S} \times \frac{d_{\mathrm{init}}}{4}},$$
$$\widetilde{\boldsymbol{Z}}_u^t = \mathrm{Concat}(\widetilde{\boldsymbol{Z}}_{u,\mathrm{N}}^t, \widetilde{\boldsymbol{Z}}_{u,\mathrm{E}}^t, \widetilde{\boldsymbol{Z}}_{u,\Phi}^t, \widetilde{\boldsymbol{Z}}_{u,\mathrm{C}}^t) \in \mathbb{R}^{\mathcal{N}_\mathrm{S} \times d_{\mathrm{init}}}, \tag{3}$$

where $\boldsymbol{W}_* \in \mathbb{R}^{d_* \times \frac{d_{\mathrm{init}}}{4}}$ and $\boldsymbol{b}_* \in \mathbb{R}^{\frac{d_{\mathrm{init}}}{4}}$ are trainable parameters. $*$ is a placeholder, which can be replaced by N, E, $\Phi$, and C.

### 4.3 TEMPORAL MOTIF EXTENSION MODULE

**Temporal Motif Extension.** The semantics of temporal motifs with different time window lengths vary significantly. To capture these various semantics, we first extend original temporal motifs based on different time window lengths. Given an original temporal motif $m$, we split its time window $\delta$ into $\mathcal{N}_\mathrm{X}$ extended time windows. Correspondingly, time window length of the $i$-th extended temporal motif $\widetilde{m}_i$ is $\widetilde{\delta}_i = i\widetilde{\delta}_\mathrm{X}$, where $\widetilde{\delta}_\mathrm{X} = \delta/\mathcal{N}_\mathrm{X}$.

**Affinity Matrix.** To construct affinity matrices that can describe the relationships between nodes and extended temporal motifs in $G_u^t$, we should perform extended temporal motif searching to enumerate the motif occurrence countings of each node, whose time complexity exponentially scales with the number of constituent egdes of temporal motifs. Fortunately, Chen et al. (2023) show motifs with more than three nodes/edges can be replaced by multiple motifs with three nodes/edges. Therefore, we only adopt temporal motifs with three edges as original temporal motifs. Figure 2 shows all temporal motifs with three edges in dynamic graphs.

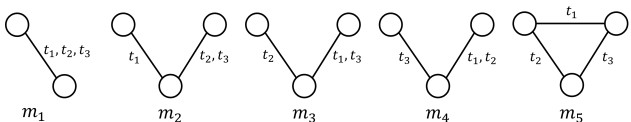

Figure 2: All temporal motifs with three edges in dynamic graphs.

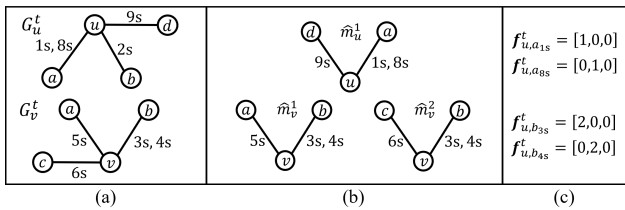

Figure 3: An example of motif occurence counting for afinity matrix.

After searching all the instances of extended temporal $\widetilde{m}$, given $x$ in $G_u^t$ with timestamp $t_x$, $\boldsymbol{f}_{u,x}^t \in \mathbb{R}^{\mathcal{N}_{\widetilde{m}}}$ is its motif occurrence countings in all the instances of $\widetilde{m}$, where $\mathcal{N}_{\widetilde{m}}$ is the number of extended constituent edges of $\widetilde{m}$ and the number in $j$-th dimension represents its occurrence counting in the $j$-th extended constituent edge.

For $m_4$ in Figure 2, its constituent edges can be formulated by $e_1 = (x_1, x_2, t_1)$, $e_2 = (x_1, x_2, t_2)$, and $e_3 = (x_1, x_3, t_3)$. Figure 3 shows an example of motif occurence counting for afinity matrix. Figure 3(a) exhibits subgraph $G_u^t$ and $G_v^t$, where we search all the instances of $m_4$. In Figure 3(b), the instance of $m_4$ in $G_u^t$ is $\widehat{m}_u^1 = \{(u, a, 1\mathrm{s}), (u, a, 8\mathrm{s}), (u, d, 9\mathrm{s})\}$ and the instances in $G_v^t$ are $\widehat{m}_v^1 = \{(v, b, 3\mathrm{s}), (v, b, 4\mathrm{s}), (v, a, 5\mathrm{s})\}$ and $\widehat{m}_v^2 = \{(v, b, 3\mathrm{s}), (v, b, 4\mathrm{s}), (v, c, 6\mathrm{s})\}$. $a_{1\mathrm{s}}$ denotes node $a$ with timestamp 1s, which only appears in the first edge of $\widehat{m}_u^1$. Therefore, in Figure 3(c), $\boldsymbol{f}_{u,a_{1\mathrm{s}}}^t = [1, 0, 0]$. Similarly, $\boldsymbol{f}_{u,a_{8\mathrm{s}}}^t = [0, 1, 0]$. $b_{3\mathrm{s}}$ appears in the first edge of $\widehat{m}_v^1$ and $\widehat{m}_v^2$ and gets $\boldsymbol{f}_{v,b_{3\mathrm{s}}}^t = [2, 0, 0]$. Similarly, $\boldsymbol{f}_{v,b_{4\mathrm{s}}}^t = [0, 2, 0]$.

We normalize $\boldsymbol{f}_{u,x}^t$ to obtain affinity vector $\boldsymbol{p}_u^t$ that describes the relationship between $x$ and $\widetilde{m}$ in $G_u^t$, which can be formulated as follows:

$$\boldsymbol{p}_{u,x}^t = \frac{\boldsymbol{f}_{u,x}^t}{\sum_{y \in G_u^t} \boldsymbol{f}_{u,y}^t} \in \mathbb{R}^{\mathcal{N}_{\widetilde{m}}}. \tag{4}$$

Then, the affinity matrix that describes the relationship between all nodes and $\widetilde{m}$ in $G_u^t$ can be denoted by $\boldsymbol{P}_u^t \in \mathbb{R}^{\mathcal{N}_\mathrm{S} \times \mathcal{N}_{\widetilde{m}}}$.

## 4.4 ATTENTIVE TEMPORAL MOTIF ENCODER

Based on affinity matrices, we capture the diverse semantics of temporal motifs by encoding and attentively aggregating extended temporal motifs with different time window lengths. We use $\boldsymbol{r}_{u,i} = \boldsymbol{P}_u^t[:, i] \in \mathbb{R}^{\mathcal{N}_\mathrm{S}}$ to represent the affinity vector of $i$-th extended constituent edge of $\widetilde{m}$. Through concatenating the affinity vectors of $i$-th extended constituent edges of $m$, we can obtain $\boldsymbol{R}_{u,i}^t \in \mathbb{R}^{\mathcal{N}_\mathrm{S} \times \mathcal{N}_\mathrm{X}}$ and compute the representations of $i$-th extended constituent edges, which can be formulated as follows:

$$\boldsymbol{\Theta}_{u,i}^t = (\boldsymbol{R}_{u,i}^t)^{\mathrm{T}} \widetilde{\boldsymbol{Z}}_u^t \boldsymbol{W}_\Theta \in \mathbb{R}^{\mathcal{N}_\mathrm{X} \times d_\mathrm{init}}, \tag{5}$$

where $\boldsymbol{W}_\Theta$ is a trainable parameter.

To take into account the preferences of $u$ to the time window lengths of different original temporal motifs, we use different preference vectors to attentively aggregate the representations of $i$-th extended constituent edges as follows:

$$\boldsymbol{q}_{u,i}^t = \mathrm{MultiheadAttention}(\boldsymbol{\psi}_m, \boldsymbol{\Theta}_{u,i}^t, \boldsymbol{\Theta}_{u,i}^t) \in \mathbb{R}^{d_\mathrm{init}}, \tag{6}$$

where $\boldsymbol{q}_{u,i}^t$ is the representation of $i$-th constituent edge of $m$ and $\boldsymbol{\psi}_m \in \mathbb{R}^{d_\mathrm{init}}$ is a parameterized preference vector for $m$. $\mathrm{MultiheadAttention}(\cdot, \cdot, \cdot)$ denotes a multi-head attention network, whose

three parameters are query matrix, key matrix and value matrix, correspondingly. Then, the representations of constituent edges of all original temporal motifs can be denoted by $\boldsymbol{Q}_u^t \in \mathbb{R}^{d_{\mathcal{N}_e} \times d_{\text{init}}}$, where $\mathcal{N}_e$ is the number of constituent edges of all original temporal motifs.

## 4.5 Dual Hypergraph Temporal Message Passing Mechanism

**Explicit Hypergraph Construction.** Previous works employ heuristic methods to construct hypergraphs. For example, Feng et al. (2019) build hypergraphs based on the co-authorship among authors in citation network. Similar to the co-authorship, we generate hyperedges for the explicit hypergraph by connecting constituent edges that co-occurr in the same motif. Then, we can obtain an incidence matrix $\boldsymbol{\mathcal{H}}_u^{t,\text{E}} \in \mathbb{R}^{\mathcal{N}_M \times d_{\mathcal{N}_e}}$ for the explicit hypergraph, where $\mathcal{N}_M$ is the number of original temporal motifs. If the $j$-th constituent edge is part of $i$-th original temporal motif, the $i$-th hyperedge should contain the $j$-th constituent edge, which is denoted by $(\boldsymbol{\mathcal{H}}_u^{t,\text{E}})_{ij} = 1$. Otherwise, $(\boldsymbol{\mathcal{H}}_u^{t,\text{E}})_{ij} = 0$.

**Implicit Hypergraph Construction.** To discover the implicit high-order correlations, we first initialize the hyperedge state $\boldsymbol{S} \in \mathbb{R}^{\mathcal{N}_h \times d_{\text{init}}}$ as trainable parameters, where $\mathcal{N}_h$ is a hyperparameter, denoting the number of implicit hyperedges. Then, we can obtain an incidence matrix $\boldsymbol{\mathcal{H}}_u^{t,\text{I}}$ for the implicit hypergraph as follows:

$$\boldsymbol{\mathcal{H}}_u^{t,\text{I}} = \text{Softmax}(\text{ReLU}(\boldsymbol{S}(\boldsymbol{Q}_u^t)^{\text{T}})) \in \mathbb{R}^{d_{\mathcal{N}_h} \times d_{\mathcal{N}_e}}, \tag{7}$$

where $\text{ReLU}(\cdot)$ and $\text{Softmax}(\cdot)$ are non-linear activation functions to eliminate weak connections and normalize the value of $\boldsymbol{\mathcal{H}}_u^{t,\text{I}}$, respectively.

**Constituent Edge to Hyperedge.** To model the complex high-order correlations among constituent edges of temporal motifs, we first perform message propagation from constituent edges to hyperedges. This process aims to extract the common features of constituent edges and derive the comprehensive representations of hyperedges, which can be formulated as:

$$\boldsymbol{\mathcal{E}}_u^{t,\text{E}} = \boldsymbol{\mathcal{H}}_u^{t,\text{E}} \boldsymbol{Q}_u^t \boldsymbol{W}_h \in \mathbb{R}^{d_{\mathcal{N}_M} \times d_{\text{init}}}, \tag{8}$$

where $\boldsymbol{\mathcal{E}}_u^{t,\text{E}}$ denotes the representations of explicit hyperedges and $\boldsymbol{W}_h \in \mathbb{R}^{d_{\text{init}} \times d_{\text{init}}}$ is a trainable parameter. Similarly, we can obtain the representations of implicit hyperedges.

**Hyperedge to Constituent Edge.** We perfrom message propagation from hyperedges to constituent edges. This process aims to model the effect of common features on constituent edges and update the representations of constituent edges, which can be formulated as:

$$\widetilde{\boldsymbol{Q}}_u^t = \text{Concat}(\boldsymbol{Q}_u^t, \boldsymbol{\mathcal{H}}_u^{t,\text{E}} \boldsymbol{\mathcal{E}}_u^{t,\text{E}}, \boldsymbol{\mathcal{H}}_u^{t,\text{I}} \boldsymbol{\mathcal{E}}_u^{t,\text{I}}) \boldsymbol{W}_q + \boldsymbol{Q}_u^t \in \mathbb{R}^{d_{\mathcal{N}_e} \times d_{\text{init}}}, \tag{9}$$

where $\widetilde{\boldsymbol{Q}}_u^t$ denotes the updated representations of constituent edges and $\boldsymbol{W}_q \in \mathbb{R}^{3d_{\text{init}} \times d_{\text{init}}}$ is a trainable parameter.

**Node Representation Update.** To propagate the effect of high-order correlations from constituent edges to nodes in $G_u^t$, we first use attention mechanism to capture the relationships between nodes and constituent edges as follows:

$$(\boldsymbol{\mathcal{R}}_u^t)_{ij} = \frac{\exp((\widetilde{\boldsymbol{Z}}_u^t)_i (\widetilde{\boldsymbol{Q}}_u^t)_j^{\text{T}})}{\sum_{k=1}^{\mathcal{N}_e} \exp((\widetilde{\boldsymbol{Z}}_u^t)_i (\widetilde{\boldsymbol{Q}}_u^t)_k^{\text{T}})}, \tag{10}$$

where $\boldsymbol{\mathcal{R}}_u^t \in \mathbb{R}^{d_S \times d_{\mathcal{N}_e}}$. Then, we can obtain the updated representations of nodes in $G_u^t$, which can be formulated as:

$$\boldsymbol{Z}_u^t = \text{GELU}(\text{LN}(\boldsymbol{\mathcal{R}}_u^t \widetilde{\boldsymbol{Q}}_u^t \boldsymbol{W}_1 + \widetilde{\boldsymbol{Z}}_u^t)) \boldsymbol{W}_2 + \widetilde{\boldsymbol{Z}}_u^t \in \mathbb{R}^{\mathcal{N}_S \times d_{\text{init}}}, \tag{11}$$

where $\boldsymbol{W}_1, \boldsymbol{W}_2 \in \mathbb{R}^{d_{\text{init}} \times d_{\text{init}}}$ are trainable parameters.

## 4.6 Aggregation and Link Predition

**Aggregation.** Finally, we follow Yu et al. (2023) to use a Transfomer encoder to aggregate the representations of nodes in $G_u^t$ and $G_v^t$ to generate the representations of $u$ and $v$, which is composed

of multi-head attention networks and feed-forwad networks. As for learning temporal dependencies between $G_u^t$ and $G_v^t$, we stack the updated representations $\boldsymbol{Z}^t = \text{Concat}(\boldsymbol{Z}_u^t, \boldsymbol{Z}_v^t) \in \mathbb{R}^{2\mathcal{N}_S \times d_{\text{init}}}$ as input, which can be formulated as follows:

$$
\begin{aligned}
\boldsymbol{O}^{t,l} &= \text{MultiheadAttention}(\boldsymbol{Z}^{t,l-1}, \boldsymbol{Z}^{t,l-1}, \boldsymbol{Z}^{t,l-1}) + \boldsymbol{Z}^{t,l-1} \in \mathbb{R}^{2\mathcal{N}_S \times d_{\text{init}}}, \\
\boldsymbol{Z}^{t,l} &= \text{FFN}^l(\text{LN}(\boldsymbol{O}^{t,l})) + \boldsymbol{Z}^{t,l-1} \in \mathbb{R}^{2\mathcal{N}_S \times d_{\text{init}}},
\end{aligned}
\tag{12}
$$

where the input of first layer is $\boldsymbol{Z}^{t,0} = \boldsymbol{Z}^t$ and the output of $L$-th layer is denoted by $\boldsymbol{Z}^{t,L} \in \mathbb{R}^{2\mathcal{N}_S \times d_{\text{init}}}$. $\text{FFN}^l(\cdot)$ is a feed-forwad network of $l$-th layer.

Then, the final representations of $u$ and $v$, denoted by $\boldsymbol{h}_u^t$ and $\boldsymbol{h}_v^t$, are calculated from $\boldsymbol{Z}^{t,L}$ by mean pooling:

$$
\begin{aligned}
\boldsymbol{h}_u^t &= \text{Mean}(\boldsymbol{Z}^{t,L}[0 : \mathcal{N}_S, :])\boldsymbol{W}_{\text{out}} + \boldsymbol{b}_{\text{out}} \in \mathbb{R}^d, \\
\boldsymbol{h}_v^t &= \text{Mean}(\boldsymbol{Z}^{t,L}[\mathcal{N}_S : 2\mathcal{N}_S, :])\boldsymbol{W}_{\text{out}} + \boldsymbol{b}_{\text{out}} \in \mathbb{R}^d,
\end{aligned}
\tag{13}
$$

where $\boldsymbol{W}_{\text{out}} \in \mathbb{R}^{d_{\text{init}} \times d}$ and $\boldsymbol{b}_{\text{out}} \in \mathbb{R}^d$ are trainable parameters.

**Link Predition.** Based on $\boldsymbol{h}_u^t$ and $\boldsymbol{h}_v^t$, the probability of an edge being generated between $u$ and $v$ at time $t$ is computed as follows:

$$
p_{u,v}^t = \text{Sigmoid}(\text{MLP}_p(\text{Concat}(\boldsymbol{h}_u^t, \boldsymbol{h}_v^t))).
\tag{14}
$$

Then, we introduce a supervised binary cross-entropy loss as the objective function.

## 5 EXPERIMENTS

### 5.1 EXPERIMENTAL SETUP

**Datasets and Baselines.** We evaluate TMMP on four real-world datasets: UCI, Enron, Wikipedia, and Reddit, which are collected by Poursafaei et al. (2022). Details of the datasets can be found in Appendix A. We perform link prediction task on these datasets and chronologically split them with the ratio of 70%/15%/15% for training/validation/testing set. Moreover, we compare TMMP with vanilla methods, i.e., JODIE (Kumar et al., 2019), DyRep (Trivedi et al., 2019), TGAT (Xu et al., 2020), TGN (Rossi et al., 2020), TCL (Wang et al., 2021a), GraphMixer (Cong et al., 2023), and RepeatMixer (Zou et al., 2024), and extra feature based methods, i.e., CAWN (Wang et al., 2021b), DyGFormer (Yu et al., 2023), and CNEN (Cheng et al., 2024). Details of the baselines can be found in Appendix B.

**Experimental Settings.** We follow Yu et al. (2023) to evaluate with two settings: transductive and inductive settings. The main objective of the transductive link prediction task is to predict possible edges between nodes that have already appeared in the training set, while the objective of the inductive link prediction task is to predict possible edges between nodes that have not appeared in the training set. Average Precision (AP) and Area Under the Receiver Operating Characteristic Curve (AUC-ROC) are employed as the evaluation metrics. The experiments are repeated five times using random seeds ranging from 0 to 4. The reported results are the average values and standard deviations of these five runs. Implementation details can be found in Appendix C. The source code of TMMP is available at `https://anonymous.4open.science/r/TMMP`.

### 5.2 PERFORMANCE COMPARISONS

We report the AP of all methods for transductive and inductive link prediction tasks in Table 1. Notably, TMMP consistently achieves the best performance, surpassing the state-of-the-art baseline methods in AP on all the datasets. It affirms the efficacy of incorporating temporal motifs into message passing for modeling the high-order correlations among nodes. Compared to the temporal motif-based method, i.e., CAWN, TMMP achieves superior performance. The reason may be that the dual hypergraph message passing mechanism can model the complex high-order correlations among the constituent edges of temporal motifs. In addition, it is obvious that extra feature based methods, i.e., CAWN, DyGFormer, CNEN, and TMMP, generally outperform vanilla methods except RepeatMixer. This is because the former methods leverage extra features to capture richer spatial and temporal information while the latter only rely on node/edge features and time encodings. RepeatMixer gets better performance than other vanilla methods through its repeat-aware

Table 1: AP of all methods for transductive and inductive link prediction tasks. The best and the second-best results are highlighted by **bold** and underlined formatting. For ease of presentation, the AP and AUC-ROC metrics in all subsequent experiments are multiplied by 100.

| Method | Transductive | | | | Inductive | | | |
|---|---|---|---|---|---|---|---|---|
| | UCI | Enron | Wikipedia | Reddit | UCI | Enron | Wikipedia | Reddit |
| JODIE (2019) | 86.70 ± 2.17 | 83.97 ± 0.42 | 96.45 ± 0.23 | 98.34 ± 0.03 | 76.61 ± 2.79 | 80.15 ± 2.29 | 94.82 ± 0.22 | 96.49 ± 0.26 |
| DyRep (2019) | 74.23 ± 4.63 | 83.06 ± 2.69 | 94.59 ± 0.43 | 98.20 ± 0.09 | 65.25 ± 3.48 | 76.60 ± 4.54 | 91.94 ± 0.62 | 95.95 ± 0.07 |
| TGAT (2020) | 85.74 ± 0.81 | 66.40 ± 1.94 | 94.97 ± 0.22 | 96.65 ± 0.03 | 83.30 ± 0.57 | 65.21 ± 1.51 | 94.66 ± 0.23 | 93.99 ± 0.18 |
| TGN (2020) | 93.39 ± 0.79 | 87.47 ± 1.27 | 98.46 ± 0.08 | 98.60 ± 0.03 | 89.67 ± 1.30 | 81.44 ± 2.11 | 97.89 ± 0.04 | 97.34 ± 0.06 |
| TCL (2021) | 92.19 ± 0.42 | 86.12 ± 0.34 | 97.84 ± 0.08 | 97.89 ± 0.01 | 90.11 ± 0.33 | 84.06 ± 0.67 | 97.59 ± 0.06 | 96.43 ± 0.07 |
| GraphMixer (2023) | 94.73 ± 0.33 | 81.94 ± 0.23 | 96.89 ± 0.03 | 97.00 ± 0.03 | 92.37 ± 0.15 | 74.92 ± 0.63 | 96.40 ± 0.07 | 94.85 ± 0.05 |
| RepeatMixer (2024) | 96.70 ± 0.15 | 92.27 ± 0.06 | 99.10 ± 0.02 | 99.12 ± 0.02 | 95.02 ± 0.18 | 88.59 ± 0.34 | 98.72 ± 0.02 | 98.69 ± 0.03 |
| CAWN (2021) | 94.91 ± 0.05 | 89.52 ± 0.13 | 98.77 ± 0.01 | 99.11 ± 0.01 | 92.39 ± 0.06 | 86.48 ± 0.30 | 98.24 ± 0.01 | 98.63 ± 0.02 |
| DyGFormer (2023) | 95.83 ± 0.05 | 92.25 ± 0.30 | 99.03 ± 0.02 | 99.21 ± 0.01 | 94.57 ± 0.09 | 89.57 ± 0.74 | 98.60 ± 0.02 | 98.82 ± 0.03 |
| CNEN (2024) | 96.01 ± 0.04 | 91.11 ± 0.04 | 98.58 ± 0.05 | 99.22 ± 0.01 | 94.15 ± 0.07 | 88.53 ± 0.12 | 98.00 ± 0.08 | 98.73 ± 0.02 |
| TMMP (Ours) | **96.86 ± 0.02** | **92.48 ± 0.16** | **99.22 ± 0.01** | **99.29 ± 0.01** | **95.54 ± 0.08** | **89.64 ± 0.16** | **98.79 ± 0.04** | **98.94 ± 0.02** |

neighbor sampling strategy. All methods exhibit relatively strong performance on Wikipedia and Reddit, owing to the informative edge features of the two datasets. The AUC-ROC of all methods for transductive and inductive link prediction tasks can be found in Appendix D, demonstrating results similar to those observed in AP. In conclusion, TMMP achieves the superior performance in both AP and AUC-ROC.

## 5.3 Ablation Study

To assess the effectiveness of the proposed modules in TMMP, we conduct an ablation study on UCI. Table 2 reports the AP and AUC-ROC of TMMP and its variants for transductive and inductive link prediction tasks, where TMMP-w/o-EXT removes the temporal motif extension, TMMP-w/o-TME removes the attentive temporal motif encoder, TMMP-w/o-EXP removes the explicit hypergraph construction, TMMP-w/o-IMP removes the implicit hypergraph construction, and TMMP-w/o-DHG removes the dual hypergraph temporal motif message passing mechanism. We can obtain the following observations:

(1) The performance degradation of TMMP-w/o-EXT and TMMP-w/o-TME relative to TMMP underscores the importance of the corporation of the temporal motif extension module and temporal motif encoder. The temporal motif extension module can generate extended temporal motifs based on different time window lengths. Then, the temporal motif encoder can encode and attentively aggregate extended temporal motifs to capture the diverse semantics of temporal motfis.

(2) The inferior results of TMMP-w/o-EXP and TMMP-w/o-IMP highlights the effectiveness of the explicit hypergraph construction and implicit hypergraph construction. TMMP-w/o-DHG shows a performance drop, emphasizing the effectiveness of the dual hypergraph temporal motif message passing mechanism to comprehensively model the complex high-order correlations.

Table 2: Performance of TMMP and its variants for transductive and inductive link prediction tasks on UCI. The best results are highlighted by **bold** formatting.

| Method | Transductive | | Inductive | |
|---|---|---|---|---|
| | AP | AUC-ROC | AP | AUC-ROC |
| TMMP-w/o-EXT | 96.60 ± 0.06 | 95.64 ± 0.08 | 95.33 ± 0.11 | 93.90 ± 0.14 |
| TMMP-w/o-TME | 96.74 ± 0.07 | 95.78 ± 0.12 | 95.48 ± 0.11 | 94.18 ± 0.10 |
| TMMP-w/o-EXP | 96.76 ± 0.07 | 95.86 ± 0.09 | 95.42 ± 0.10 | 94.02 ± 0.12 |
| TMMP-w/o-IMP | 96.79 ± 0.09 | 95.92 ± 0.10 | 95.45 ± 0.14 | 94.15 ± 0.18 |
| TMMP-w/o-DHG | 96.72 ± 0.06 | 95.89 ± 0.08 | 95.41 ± 0.10 | 94.10 ± 0.18 |
| TMMP | **96.86 ± 0.02** | **96.01 ± 0.02** | **95.54 ± 0.08** | **94.25 ± 0.13** |

## 5.4 Hyperparameter Sensitivity Analysis

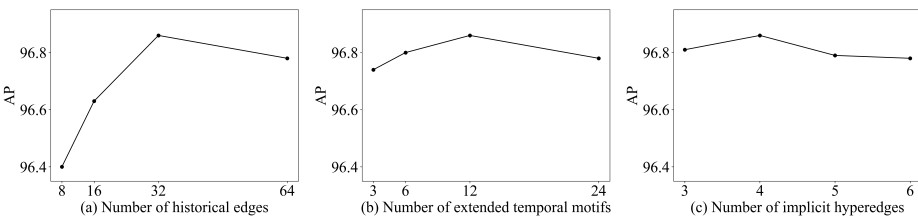

Figure 4: AP of TMMP with different hyperparameters for transductive link prediction task on UCI.

Table 3: Performance of extra feature based methods for transductive link prediction task on UCI across different numbers of common neighbors. #NC denotes the number of common neighbors. The percentage values in parentheses indicate the proportion of the corresponding number of common neighbors in the testing set. Other notations follow Table 2.

| Method | #NC=0 (52.40%) | | #NC=1 (8.50%) | | #NC=2 (32.47%) | | #NC>2 (6.63%) | |
|---|---|---|---|---|---|---|---|---|
| | AP | AUC-ROC | AP | AUC-ROC | AP | AUC-ROC | AP | AUC-ROC |
| CAWN | 40.48 ± 0.54 | 77.34 ± 0.21 | 80.54 ± 0.34 | 82.71 ± 0.21 | 99.71 ± 0.02 | 89.97 ± 0.34 | 99.67 ± 0.02 | 90.57 ± 0.34 |
| DyGFormer | 43.63 ± 0.66 | 78.74 ± 0.24 | 82.25 ± 0.67 | 83.74 ± 0.70 | 99.82 ± 0.01 | 95.36 ± 0.34 | 99.75 ± 0.03 | 93.25 ± 0.72 |
| CNEN | 45.77 ± 0.47 | 76.87 ± 0.22 | 81.92 ± 0.17 | 84.00 ± 0.12 | 99.83 ± 0.02 | 95.68 ± 0.16 | **99.79 ± 0.02** | **94.90 ± 0.40** |
| TMMP | **54.56 ± 0.95** | **83.27 ± 0.21** | **85.40 ± 0.11** | **87.40 ± 0.18** | **99.87 ± 0.01** | **95.95 ± 0.10** | 99.71 ± 0.08 | 91.98 ± 0.90 |

We study the hyperparameter sensitivity of TMMP, including the number of historical edges ($\mathcal{N}_\text{S}$), the number of extended temporal motifs ($\mathcal{N}_\text{X}$), and the number of implicit hyperedges ($\mathcal{N}_\text{h}$). The impact of these hyperparameters is shown in Figure 4. We can observe that: (1) The optimal number of historical edges is 32 in Figure 4(a). The reason is that the smaller number cannot provide sufficient spatial and temporal information and the larger number may introduce outdated information. (2) The optimal number of extended temporal motifs is 12 in Figure 4(b). The reason is that the smaller number cannot thoroughly capture the diverse semantics of temporal motifs and the larger number may raise the risk of overfitting. (3) The optimal number of implicit hyperedges is 4 in Figure 4(c). The reason is that the smaller number cannot comprehensively model the implicit high-order correlations and the larger number may introduce noise.

### 5.5 Influence of Different Numbers of Common Neighbors

The presence of common neighbors between two target nodes often indicates a high likelihood of a future edge being generated between them. To explore the influence of different numbers of common neighbors on extra feature based methods, we partition the testing set according to the number of common neighbors. Table 3 shows the AP and AUC-ROC of these methods for transductive link prediction task on UCI. We can obtain the following observations: (1) all methods achieve strong performance when common neighbors are present, highlighting the importance of common neighbors in link prediction tasks. (2) TMMP perform slightly worse than DyGFormer and CNEN when there are more than 2 common neighbors, because the two methods can effectively utilize more common neighbors. (3) TMMP outperforms other methods when there are fewer common neighbors. This advantage stems from its ability to capture the diverse semantics of temporal motifs from subgraphs of target nodes, which are more dependent on subgraph structures than on common neighbors, thereby compensating for the lack of common neighbors.

### 5.6 Computational Cost

We compare TMMP with extra feature based methods, i.e., CAWN, DyGFormer, and CNEN, on UCI with 32 historical edges. To accelerate the training of TMMP, we first preprocess the data to perform temporal motif searching. Table 4 reports the training time of these methods per epoch, inference time in validation set, and the number of parameters. Considering the performance improvement in the absence of common neighbors, which is frequently encountered in the real world, and the reasonable training and inference time, TMMP demonstrates its superiority over other methods. The complexity analysis of TMMP can be found in Appendix E.

Table 4: Computational cost on UCI.

| Method | CAWN | DyGFormer | CNEN | TMMP |
|---|---|---|---|---|
| Training Time | 34.24s | 11.05s | 2.27s | 16.16s |
| Inference Time | 6.91s | 1.78s | 0.28s | 2.32s |
| #Parameters | 15.35MB | 4.15MB | 0.76MB | 10.28MB |

## 6 Conclusions

In this paper, we propose TMMP to incorporate temporal motifs into message passing networks, which can model the high-order correlations among nodes. Empowered by the temporal motif extension module and attentive temporal motif encoder, TMMP can capture the diverse semantics of temporal motfis. With the dual hypergraph temporal motif message passing mechanism, TMMP can comprehensively model the complex high-order correlations. Experiments are conducted on four real-world datasets, which demonstrate the superior performance of TMMP.

## 7 ETHICS STATEMENT

All authors have read and adhere to the ICLR Code of Ethics. This work uses publicly available benchmarks and introduces no new data collection. It involves no human subjects, no personally identifiable information, and no sensitive attributes. There are no conflicts of interest. We do not identify any ethical concerns specific to this submission.

## 8 REPRODUCIBILITY STATEMENT

The source code of TMMP is available at `https://anonymous.4open.science/r/TMMP`. Hyperparameters and protocol details are described in the paper and the appendix to facilitate end-to-end reproduction.

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

Table 5: Statics of the datasets.

| Datasets | Nodes | Edges | Node/Edge Feature | Unique Pairs | Unique Timestamps | Duration |
|---|---|---|---|---|---|---|
| UCI | 1,899 | 59,835 | -/- | 20,296 | 58,911 | 196 days |
| Enron | 184 | 125,235 | -/- | 3,125 | 22,632 | 3 years |
| Wikipedia | 9,227 | 157,474 | -/172 | 18,257 | 152,757 | 1 month |
| Reddit | 10,984 | 672,447 | -/172 | 78,516 | 669,065 | 1 month |

## A  DESCRIPTION OF DATASETS

We evaluate the performance of various dynamic graph representation learning methods on four real-world datasets collected and publicly released by Poursafaei et al. (2022). Among them, UCI and Enron are homogeneous graphs representing user-user interactions, while Wikipedia and Reddit are bipartite graphs capturing user-item interactions. Below are the dataset specifics:

- **UCI** (also known as CollegeMsg) is an online communication network among students at the University of California, Irvine, spanning 196 days with timestamps precise to the second. This dataset does not have node or edge features and consists of 1,899 student nodes, 59,835 edges (records), and 58,911 distinct timestamps. Each record represents the behavior of a student sending a message to another student at a specific time. Without considering timestamps, there are 20,296 distinct node pairs in this dataset.

- **Enron** is an email network among employees of the Enron Corporation, spanning three years with timestamps precise to the second. This dataset does not have node or edge features and consists of 184 employee nodes, 125,235 edges (records), and 22,632 distinct timestamps. Each record represents the behavior of an employee sending an email to another employee at a specific time. Without considering timestamps, there are 3,125 distinct node pairs in this dataset.

- **Wikipedia** is a user-edit interaction network for Wikipedia articles, spanning one month with timestamps precise to the second. This dataset does not have node features, but each record is associated with a 172-dimensional Linguistic Inquiry and Word Count (LIWC) feature. It consists of 8,277 user nodes, 1,000 article nodes, 157,474 edges (records), and 152,757 distinct timestamps. Each record represents the behavior of a user editing a Wikipedia article at a specific time. Without considering timestamps, there are 18,257 distinct node pairs in this dataset.

- **Reddit** is a user-subreddit interaction network from the Reddit online community, spanning one month with timestamps precise to the second. This dataset does not have node features, but each record is associated with a 172-dimensional LIWC feature. It consists of 9,984 user nodes, 1,000 subreddit nodes, 672,447 edges (records), and 669,065 distinct timestamps. Each record represents the behavior of a user posting in a subreddit at a specific time. Without considering timestamps, there are 78,516 distinct node pairs in this dataset.

The statistical characteristics of each dataset are shown in Table 5.

## B  DESCRIPTION OF BASELINES

The following 10 baselines are selected, which are categoried into vanilla methods and extra feature based methods. Details of vanilla methods are as follows:

- **JODIE** (Kumar et al., 2019) employs dual recurrent neural networks to update user and item representations, while using a temporal projection function to model the evolution of node representations over time.

- **DyRep** (Trivedi et al., 2019) adopts recurrent neural networks to update node states on each interaction and applys a temporal-attentive aggregation module to model the temporal dynamics.

- **TGAT** (Xu et al., 2020) utilizes a time encoding function to model temporal patterns, aggregates temporal neighbor information via an attention network, and captures higher-order neighborhood information through stacked layers.

- **TGN** (Rossi et al., 2020) introduces a recurrent neural network to update node memories in chronological order, samples a fixed number of temporal neighbors to reduce computational overhead, and employs an attention mechanism to aggregate their information.

- **TCL** (Wang et al., 2021a) retrieves node interaction sequences via breadth-first search in a temporal dependency interaction graph, extends the original Transformer encoder to handle spatial and temporal structures in sequences, and models dependencies between interacting nodes via cross-attention.
- **GraphMixer** (Cong et al., 2023) replaces parameterized time encoding with a parameter-free time encoding and leverages MLP-Mixer to capture spatial and temporal information from temporal neighbors.
- **RepeatMixer** (Zou et al., 2024) introduces a repeat-aware neighbor sampling strategy to utilize the recent neighbors that occurr before the neighbors historically interating with the target nodes.

Details of extra feature based methods are as follows:

- **CAWN** (Wang et al., 2021b) introduces causal anonymous walks to extract instances of temporal motif. Then, it uses recurrent neural networks to encode each walk and aggregate them to generate node representations.
- **DyGFormer** (Yu et al., 2023) introduces neighbor co-occurrence encodings to capture the relationship between the first-order temporal neighbor sequences of two target nodes. Leveraging a Transformer encoder and patch techniques, it efficiently extracts spatial and temporal structural information from longer interaction histories.
- **CNEN** (Cheng et al., 2024) adopts hashtable-based memories and GPU acceleration to compute co-neighbor encodings, which can count common neighbors with high efficiency. Moreover, it generates long-term and short-term co-neighbor encodings to capture the temporal patterns.

## C  IMPLEMENTATION DETAILS

TMMP is implemented based on the PyTorch 1.12.0 framework and Python 3.10. All experiments are conducted on an NVIDIA RTX 3080 GPU. The Adam optimizer is adopted with an initial learning rate of 0.0001. During training, the batch size and dropout rate are set to 200 and 0.1. The best-performing model on the validation set is selected for testing. Due to high computational complexity, we accelerate the training by setting the maximum number of training epochs to 50 and employing an early stopping strategy with a patience of 10. To be consistent, we rerun all the baselines following their official hyperparameter configurations.

For the hyperparameters of TMMP, we set time encoding dimension ($d_\Phi$) to 100, subgraph co-occurrence encoding dimension ($d_C$) to 50, initialized representation dimension ($d_{\text{init}}$) to 200, final representation dimension ($d$) to 172, number of attention heads to 2, time window length of original temporal motifs ($\delta$) to 12 days, number of extended temporal motifs ($\mathcal{N}_X$) to 12, and Transformer layer ($L$) to 2. We set TMMP layer to 2 for Reddit and 1 for other datasets. For number of historical edges ($\mathcal{N}_S$), we perform a grid search with search space $\{8, 16, 32, 64\}$ and find the final settings of 32, 64, 64, and 64 for UCI, Enron, Wikipedia, and Reddit, respectively. For number of implicit hyperedges ($\mathcal{N}_h$), we perform a grid search with search space $\{3, 4, 5, 6\}$ and find the final settings of 4, 3, 6, and 4 for UCI, Enron, Wikipedia, and Reddit, respectively.

## D  PERFORMANCE COMPARISONS IN AUC-ROC

We report the AUC-ROC of all methods for transductive and inductive link prediction tasks in Table 6. It is obvious that TMMP achieves the superior performance, outperforming the state-of-the-art baseline methods in AUC-ROC on three of the four datasets. These results confirm the effectiveness of incorporating temporal motifs into message passing for modeling the high-order correlations among nodes. TMMP fails to achieve the best performance on Enron, which is characterized by a larger number of common neighbors between two target nodes. Under this condition, the advantage of TMMP is relatively limited (more details can be found in Section 5.5).

## E  COMPLEXITY ANALYSIS

For the subgraph encoding module, the time complexity of encoding subgraph co-occurrence is $\mathcal{O}(n\mathcal{N}_S^2)$, where $n$ is the number of edges in datasets, and $\mathcal{N}_S$ is the number of historical edges.

Table 6: AUC-ROC of all methods for transductive and inductive link prediction tasks. The notations follow Table 1.

| Method | Transductive | | | | Inductive | | | |
|---|---|---|---|---|---|---|---|---|
| | UCI | Enron | Wikipedia | Reddit | UCI | Enron | Wikipedia | Reddit |
| JODIE (2019) | 89.18 ± 1.03 | 87.25 ± 0.21 | 96.27 ± 0.16 | 98.30 ± 0.02 | 77.14 ± 1.43 | 81.23 ± 1.94 | 94.28 ± 0.25 | 96.48 ± 0.16 |
| DyRep (2019) | 76.79 ± 4.76 | 83.88 ± 2.13 | 94.12 ± 0.49 | 98.16 ± 0.08 | 65.52 ± 4.26 | 78.11 ± 3.85 | 90.97 ± 0.77 | 95.93 ± 0.05 |
| TGAT (2020) | 85.30 ± 0.74 | 64.20 ± 2.17 | 94.04 ± 0.25 | 96.43 ± 0.03 | 82.01 ± 0.60 | 61.96 ± 1.21 | 93.72 ± 0.25 | 93.61 ± 0.17 |
| TGN (2020) | 93.20 ± 0.80 | 89.18 ± 1.10 | 98.37 ± 0.08 | 98.56 ± 0.04 | 88.47 ± 1.50 | 83.98 ± 2.78 | 97.78 ± 0.05 | 97.23 ± 0.08 |
| TCL (2021) | 91.35 ± 0.48 | 84.08 ± 0.35 | 97.42 ± 0.10 | 97.65 ± 0.02 | 88.53 ± 0.45 | 82.24 ± 0.85 | 97.17 ± 0.09 | 95.95 ± 0.09 |
| GraphMixer (2023) | 93.85 ± 0.45 | 84.33 ± 0.17 | 96.55 ± 0.04 | 96.88 ± 0.02 | 91.06 ± 0.17 | 75.86 ± 0.91 | 96.04 ± 0.07 | 94.61 ± 0.05 |
| RepeatMixer (2024) | 95.38 ± 0.18 | **93.25 ± 0.11** | 99.00 ± 0.02 | 99.02 ± 0.02 | 93.58 ± 0.25 | 89.35 ± 0.33 | 98.61 ± 0.02 | 98.50 ± 0.04 |
| CAWN (2021) | 93.84 ± 0.05 | 90.36 ± 0.27 | 98.55 ± 0.02 | 99.02 ± 0.01 | 90.25 ± 0.04 | 87.09 ± 0.48 | 98.04 ± 0.03 | 98.43 ± 0.03 |
| DyGFormer (2023) | 94.45 ± 0.05 | 93.04 ± 0.25 | 98.91 ± 0.01 | 99.13 ± 0.01 | 92.67 ± 0.14 | **90.44 ± 0.51** | 98.50 ± 0.02 | 98.68 ± 0.03 |
| CNEN (2024) | 94.71 ± 0.08 | 91.86 ± 0.07 | 98.34 ± 0.05 | 99.16 ± 0.01 | 91.85 ± 0.11 | 88.16 ± 0.14 | 97.71 ± 0.10 | 98.60 ± 0.01 |
| TMMP (Ours) | **96.01 ± 0.02** | 92.98 ± 0.16 | **99.17 ± 0.01** | **99.24 ± 0.01** | **94.25 ± 0.13** | 90.16 ± 0.26 | **98.74 ± 0.03** | **98.83 ± 0.02** |

For the temporal motif extension module, the time complexity of temporal motif searching is $\mathcal{O}(n\mathcal{N}_{\mathrm{S}}^{\mathcal{N}_m})$, where $\mathcal{N}_m$ is the number of edges of temporal motifs. For the attentive temporal motif encoder, the time complexity is $\mathcal{O}(n\mathcal{N}_{\mathrm{X}})$, where $\mathcal{N}_{\mathrm{X}}$ is the number of extended time windows. For the dual hypergraph temporal message passing mechanism, the time complexity is $\mathcal{O}(n(\mathcal{N}_{\mathrm{M}}+\mathcal{N}_{\mathrm{h}}))$, where $\mathcal{N}_{\mathrm{M}}$ is the number of temporal motifs and $\mathcal{N}_{\mathrm{h}}$ is the number of implicit hyperedges.

# F    THE USE OF LARGE LANGUAGE MODELS

We used a large language model (ChatGPT) only to aid writing, including grammar correction and minor phrasing edits. All suggestions were reviewed and edited by the authors, who take full responsibility for the content.

