# OpenReview forum: "Temporal Motif Message Passing Network for Dynamic Graph Representation Learning"
_ICLR.cc/2026/Conference — ICLR 2026 Conference Withdrawn Submission_

### Official Review · Reviewer_LFzg · 2025-10-25

**Soundness:** 2
**Presentation:** 2
**Contribution:** 2
**Rating:** 2
**Confidence:** 4

**Summary:**

To effectively capture high-order and complex relationships among nodes, this paper proposes a motif-based message passing extension for transformer-based dynamic graph learning methods. For each target node u and its historical edges, the method first constructs affinity matrices that record the frequency of each edge appearing within a set of predefined motifs over multiple time intervals. These matrices are then leveraged to update the representations of u’s historical edges using a hypergraph temporal message passing mechanism. Finally, transformer models are applied to the updated edge representations to generate the final embedding for node u.

**Strengths:**

- The observation about the diverse semantics of temporal motifs make senses, and the ability to effectively handle such diversity could be crucial for applications such as fraud detection.
- Tackling the diverse semantics of temporal motifs by varying their time intervals is a natural and valid solution.
- The proposed method achieves better performance than the compared baselines.

**Weaknesses:**

- The complexity of the proposed method is high, which might limit its application to small temporal graphs. Besides, such high complexity (in both concepts and computation) is also doubtful. This method builds upon already complicated sequence-based Transformers by adding motif affinity matrix construction, additional attention mechanisms, and a hypergraph neural network. However, as shown in Table 2, the performance improvement is marginal, while the computational cost—especially from motif matching—is extremely high. The authors should reconsider whether such design complexity is truly necessary.
- The choice of datasets is limited and somewhat inappropriate. Most of the selected datasets exhibit weak higher-order patterns, making them unsuitable for evaluating a method designed to capture complex high-order dependencies. For example, the Wikipedia dataset is dominated by repetitive behaviors rather than intricate structural motifs. It would be more convincing to include datasets with richer higher-order patterns, such as MOOC, LastFM and some financial fraud detection datasets.

**Questions:**

Please see the Weaknesses.

---

### Official Review · Reviewer_3cQC · 2025-10-26

**Soundness:** 2
**Presentation:** 3
**Contribution:** 2
**Rating:** 4
**Confidence:** 4

**Summary:**

This paper proposes a framework that incorporates temporal motifs into message passing networks for dynamic graph representation learning. The approach includes: (1) a temporal motif extension module to capture motifs at different time window lengths, (2) an attentive temporal motif encoder to capture diverse semantics, and (3) a dual hypergraph message passing mechanism (explicit and implicit hypergraphs) to model high-order correlations among constituent edges. Experiments on four datasets demonstrate state-of-the-art performance in link prediction tasks.

**Strengths:**

- The paper presents the first work to incorporate temporal motifs into message passing networks rather than just using them as auxiliary features, which is a meaningful contribution to modeling high-order correlations.
- The combination of explicit hypergraph construction and implicit hypergraph construction is innovative and comprehensive for capturing both obvious and hidden relationships among edges.
- The paper includes comprehensive ablation studies, hyperparameter sensitivity analysis, and computational cost comparisons, providing good insights into model behavior.

**Weaknesses:**

- Evaluation limited to single task type: this paper only evaluates on link prediction, although it contains transductive and inductive variants.  However, many of the baselines compared in the experiments have been evaluated on different tasks. For example, JODIE uses the user state change prediction, and TGAT chooses the dynamic node classification task.
- The experimental evaluation uses relatively small-scale datasets where UCI has only 1,899 nodes and 59,835 edges, Enron has only 184 nodes and 125,235 edges, and even the larger datasets like Wikipedia and Reddit contain only ~10k nodes. This raises concerns about scalability to real-world applications. In comparison, TGAT has been tested on Industrial data, which has 170k nodes and 2m edges.
- The performance gains over the best baselines are often very small shown in Table 1. This may raise the cost-benefit trade-off corncern, especially considering the complexity of the designed components such as motif searching and extension.
- The ablation study shows that different components have no (?) much impact on the performance. More investigations should be conducted to explain where the improvement comes from.

**Questions:**

- More diverse evaluation tasks
- Experiments on larger-scale graphs
- Explanations on ablation studies
- Discussions on cost-benefit trade-off

---

### Official Review · Reviewer_AJii · 2025-10-30

**Soundness:** 2
**Presentation:** 3
**Contribution:** 2
**Rating:** 4
**Confidence:** 4

**Summary:**

This paper introduces TMMP, a Temporal Motif Message Passing network designed to improve continuous-time dynamic graph representation learning by incorporating temporal motifs into message passing. It employs a temporal-motif extension module, an attentive motif encoder, and a dual-hypergraph message-passing mechanism to capture diverse temporal semantics and high-order correlations among motif edges.

**Strengths:**

- The paper provides a formulation and modular framework.

- Experimental results are extensive.

- The method captures high-order temporal dependencies in a principled manner.

**Weaknesses:**

- The computational complexity of motif searching and dual-hypergraph construction is very high, which may limit scalability to large-scale dynamic networks.
﻿
﻿
- The notation in the methodology section is heavy and sometimes inconsistent, reducing readability.
﻿
- Baseline coverage omits several recent 2024–2025 CTDG models and contrastive-learning approaches, which could affect fairness of comparison.
﻿
- The practical efficiency analysis reports per-epoch times but does not include end-to-end runtime or memory usage on larger datasets, making it difficult to assess deployment feasibility.
﻿
﻿
- TMMP introduces a dual hypergraph message passing mechanism to model high-order correlations. Could the authors clarify whether this mechanism captures new relational patterns that cannot be represented by existing hypergraph-based or attention-based models?
﻿
- The proposed framework stacks multiple attention layers (in motif encoding and final aggregation). Has the impact of attention depth or head number been systematically analyzed, and how does this multi-level attention influence representation interpretability or model stability?

**Questions:**

Please see weaknesses.

---

### Official Review · Reviewer_unnx · 2025-10-31

**Soundness:** 2
**Presentation:** 2
**Contribution:** 1
**Rating:** 2
**Confidence:** 4

**Summary:**

This paper proposes a new model for dynamic graph representation learning, called the Temporal Motif Message Passing Network (TMMP). The method constructs high-order dependencies through temporal motifs, transforming dynamic graphs into explicit and implicit hypergraphs, followed by multi-stage message passing. The authors validate the model on several public datasets, demonstrating performance superior to existing methods.

**Strengths:**

1. The design from temporal motif construction to the message propagation module follows a consistent logic and can be seamlessly integrated with existing dynamic graph representation learning frameworks.

**Weaknesses:**

1. TMMP introduces multiple modules (such as motif expansion, dual hypergraph construction, and attention-based aggregation), which make the model structure complex and significantly increase the number of parameters. However, the resulting performance improvement is limited. As shown in the experiments in Section 5.6, TMMP has more than ten the number of parameters of CNEN and both its training and inference times are significantly higher. Moreover, the comparison in that table is not entirely fair. To my knowledge, model such as RepeatMixer also has far fewer parameters and run much faster,, while achieving very similar performance. Yet its result is not included.

2. One of the core components of the model is the learning of the implicit hypergraph. However, the paper does not demonstrate whether the learned implicit hyperedges capture meaningful high-order structures, nor does it provide any visualization or statistical analysis. The learning mechanism of this component lacks intuitive explanation.

3. The proposed model is primarily validated through empirical experiments, with limited theoretical justification or analysis of computational complexity. Motif enumeration and expansion are theoretically exponential in complexity. While the authors claim preprocessing is feasible, no large-scale experiments are provided. For million-edge graphs, the computational cost may be prohibitive. Furthermore, the paper does not provide a clear rationale for why the specific combination of explicit and implicit hypergraph propagation should be inherently superior to alternative forms of high-order modeling in the context of temporal motifs.

4. The ablation/hyper-parameter studies lack cross-dataset consistency analysis and is limited to a single dataset (UCI). Moreover, the paper does not evaluate the model’s robustness, such as its sensitivity to time window selection or motif size.

5. The main experiments do not align with established protocols in the literature (e.g., DyGFormer). Only a single negative sampling strategy is adopted, whereas recent works typically report results under random，historical，inductive negative sampling settings. Moreover, several standard continuous-time dynamic graph benchmarks—such as MOOC, Social Evo, Flights, UNtrade, and Contact—are omitted, limiting the comparability of the results.

**Questions:**

please see the weakness above.

---

### Note · Authors · 2026-01-08

I have read and agree with the venue's withdrawal policy on behalf of myself and my co-authors.